# Molecular chlamydia and gonorrhoea point of care tests implemented into routine practice: Systematic review and value proposition development

**Sebastian S. Fuller** [1,2] *, **Eleanor Clarke** [3], **Emma M. Harding-Esch** [1,3]

**1** Institute for Infection and Immunity, Applied Diagnostic Research and Evaluation Unit, St George's University of London, London, United Kingdom, **2** Nuffield Department of Medicine, Health Systems Collaborative, University of Oxford, Headington, Oxford, United Kingdom, **3** Clinical Research Department, Faculty of Infectious and Tropical Diseases, London School of Hygiene & Tropical Medicine, London, United Kingdom

* sebastian.fuller@ndm.ox.ac.uk

**Data Availability Statement:** All relevant data are within the manuscript and its Supporting information files.

## Abstract

### Background

Sexually Transmitted Infections, including *Neisseria gonorrhoeae* (NG) and *Chlamydia trachomatis* (CT), continue to be a global health problem. Increased access to point-of-care-tests (POCTs) could help detect infection and lead to appropriate management of cases and contacts, reducing transmission and development of reproductive health sequelae. Yet diagnostics with good clinical effectiveness evidence can fail to be implemented into routine care. Here we assess values beyond clinical effectiveness for molecular CT/NG POCTs implemented across diverse routine practice settings.

### Methods

We conducted a systematic review of peer-reviewed primary research and conference abstract publications in Medline and Embase reporting on molecular CT/NG POCT implementation in routine clinical practice until 16th February 2021. Results were extracted into EndNote software and initially screened by title and abstract by one author according to the inclusion and exclusion criteria. Articles that met the criteria, or were unclear, were included for full-text assessment by all authors. Results were synthesised to assess the tests against guidance criteria and develop a CT/NG POCT value proposition for multiple stakeholders and settings.

### Findings

The systematic review search returned 440 articles; 28 were included overall. The Cepheid CT/NG GeneXpert was the only molecular CT/NG POCT implemented and evaluated in routine practice. It did not fulfil all test guidance criteria, however, studies of test implementation showed multiple values for test use across various healthcare settings and locations.

**Funding:** The authors received no specific funding for this work.

**Competing interests:** We have read the journal's policy and the authors of this manuscript have the following competing interests: ADREU has received funding from Abbott, binx health, Cepheid, SpeedDx, Mologic, Revolugen and Sekisui, for the research and evaluation of their diagnostics. SSF was a co-investigator on Innovate UK grant to binx health, "A stratified medicine diagnostic test for STI patients at the point-of-care" (ref: 971543) and is a consultant for the WHO on point-of-care tests for sexually transmitted infections. EHE and EC report that no competing interests exist. This does not alter our adherence to PLOS ONE policies on sharing data and materials.

**Abbreviations:** CE, *Conformité Européene*; CT, *Chlamydia trachomatis*; ED, Emergency Department; FDA, Food and Drug Administration; GP, General Practice; GPS, Global Positioning System; LMICs, Low- and Middle-Income Countries; NG, *Neisseria gonorrhoeae*; POCT, Point-of-Care-Test; STI, Sexually Transmitted Infection; TPP, Target Product Profile; TV, *Trichomonas vaginalis*; WHO, World Health Organization.

Our value proposition highlights that the majority of values are setting-specific. Sexual health services and outreach services have the least overlap, with General Practice and other non-sexual health specialist services serving as a "bridge" between the two.

## Conclusions

Those wishing to improve CT/NG diagnosis should be supported to identify the values most relevant to their settings and context, and prioritise implementation of tests that are most closely aligned with those values.

## Introduction

It is estimated that there are over 1 million new curable sexually transmitted infections (STI) cases every day; in 2016 there were approximately 376 million new cases of the most common curable STIs: *Neisseria gonorrhoeae* (NG), *Chlamydia trachomatis* (CT) *Trichomonas vaginalis* (TV), and syphilis [1]. If left untreated, these infections can result in serious reproductive health sequelae, such as infertility, chronic pelvic pain, ectopic pregnancy, and pelvic inflammatory disease. CT and NG infections are two key STIs: CT is the most commonly reported STI [2], and treatment of NG is a global public health problem following the emergence of multi-drug resistant strains [3].

Syndromic management (diagnosis and treatment of STIs based on patients' clinical history and reported and observed symptoms) has been shown to be both poorly sensitive and specific for STI diagnosis [4]. It can result in asymptomatic but infected individuals not being treated, resulting in continued transmission and development of reproductive health sequelae. Conversely, symptomatic patients of unknown aetiology may receive unnecessary, inappropriate and/or sub-optimal treatment, potentially increasing the risk of STI antimicrobial resistance (AMR) emergence [5]. STI diagnosis is therefore ideally informed by diagnostic tests, and there has been a marked move away from syndromic management, wherever possible, in the majority of high-income settings [6]. However, in LMICs, syndromic management is still commonplace. There is little access to large-scale laboratories, as well as a lack of highly skilled healthcare professionals and specialised equipment in clinical settings, which are needed for aetiological diagnosis of STIs [4, 5].

Diagnostics have been hailed as a critical intervention to reduce the global burden of AMR [3], with a growing need for the development of point-of-care tests (POCTs) to combat the global STI health burden [6, 7]. The World Health Organization (WHO) defines POCTs as those that can be used at, or near, the point of patient care [8]. Guidelines and criteria for optimal diagnostics have been published to both guide test development and assess their ability to meet STI control requirements in all settings [9–12]. These include the REASSURED criteria (**A**ffordable, **S**ensitive, **S**pecific, **U**ser-friendly, **R**apid and robust, **E**quipment-free and **D**eliverable to end-users, recently updated to also include advances in m-health, incorporating both "**R**eal-time connectivity" and "**E**ase of specimen collection") [9, 11]. These criteria focus on the needs of LMICs and were developed by WHO's STD Diagnostics Initiative as a benchmark to determine whether POCTs for community level (level 1 health centres) use meet local requirements for STI prevention, control and management [13]. Furthermore, POCT Target Product Profiles (TPPs) for specific infections have been created by WHO through consultation with experts. These TPPs focus both on LMIC and higher-income country needs, and include

multiple minimal and optimal characteristics, from diagnostic accuracy characteristics to cost [12]. TPPs aim to help accelerate and guide future STI POCT development [12].

There are various POCTs for CT and NG diagnosis available [8, 14, 15]. Non-molecular testing for NG includes Gram stain microscopy, which requires specialist equipment and a high-level of training of healthcare professionals within the clinic [16]. For CT, commercial antigen detection lateral flow tests have been developed with the ASSURED criteria in mind: they are low-cost, equipment-free and easy to perform, but offer suboptimal sensitivity for diagnosis [12, 17]. International guidelines stipulate that diagnosis of CT and NG should be based on results from highly-accurate molecular tests wherever possible [18–21]. To date, two highly accurate molecular POCTs for CT and NG have obtained *Conformité Européene* (CE) marking from the European Union, and United States Food and Drug Association (FDA) regulatory approval, both of which offer CT/NG dual detection: the Cepheid GeneXpert, with a 90-minute time-to-results [22], and the binx health io with a 30-minute time-to-results [23].

However, even diagnostics with excellent clinical trial outcomes face multiple barriers to adoption [24, 25]. Although TPP and REASSURED criteria are useful frameworks for test development and evaluation, different values for adoption, such as clinical, process and financial outcomes, are negotiated during implementation [26]. It is increasingly recognised that the social and structural context of implementing a new technology is as important as evidence for its clinical effectiveness [27–29], and that these should be reviewed from the different perspectives of multiple stakeholders [24, 30, 31]. Stakeholders are defined as any person or organisation contributing to a care pathway, including patients, carers, healthcare professionals, provider organisations, purchasers of healthcare services, policymakers and laboratory medicine specialists [32].

The value of POCTs is likely to differ both within and between different stakeholder groups, who often have varying priorities and objectives [33]. It is important to understand these values to facilitate the integration of POCTs into sexual healthcare. There are many proposed frameworks to measure value [34], one of which is the value proposition of laboratory medicine [35]. It aims to facilitate the implementation of innovations in healthcare by consolidating and making visible the available evidence of the innovations' costs and benefits to different stakeholders [35]. It also considers values beyond clinical trial data, arguing that in an outcomes-based health system, the value of an innovation to *all* stakeholders must be measured and communicated [32, 35, 36].

We aimed to develop a value proposition for molecular CT/NG POCTs that is reflective of the needs of different sexual healthcare stakeholders, in order to facilitate decision-making processes for implementation and adoption of CT/NG POCTs into diverse care settings.

## Methods

The overall research question was: "What are the outcomes of molecular CT/NG POCTs implementation for patients being tested for CT/NG in different routine practice settings?" To answer this question, we developed three specific objectives: i.) What values are placed on CT/NG POCTs implemented in routine practice in the published literature? ii.) Do molecular CT/NG POCTs implemented in routine practice fulfil the (RE)ASSURED and TPP criteria? iii.) What is the value proposition for molecular CT/NG POCTs by setting, based on the value proposition for laboratory medicine [35]?

To meet our first objective, we conducted a systematic review of the published literature reporting on molecular CT/NG POCT implementation in routine clinical care. To meet the second objective, we reviewed and assessed compliance of the tests identified through the systematic review to REASSURED and TTP criteria for STI POCT development, using data from

formal diagnostic evaluations. For the third objective, we developed a CT/NG POCT value proposition based on a synthesis of the wider available evidence. Data from additional studies evaluating NAAT-based test(s) for CT/NG, but that were ineligible for the systematic review (e.g. research-only outcome, such as diagnostic accuracy studies; or not primary research, such as cost-effectiveness modelling), were extracted. These were applied to the value proposition for laboratory medicine framework [35] to develop a value proposition for molecular CT/NG POCTs, by setting type. Data were tabulated to meet each objective, and a narrative synthesis of results (i.e., rather than a metanalysis) was conducted by SSF and EMHE, given the heterogeneity of study designs and settings.

The systematic review was conducted in MEDLINE and Embase (OvidSP interfaces) to include all peer-reviewed primary research and conference abstract publications until 16[th] February 2021 (S1 and S2 Tables). Both MEDLINE and Embase (OvidSP interfaces) were searched by one researcher (EC) for studies involving human participants using a combination of terms and synonyms based on four key concepts (chlamydia AND gonorrhoea AND point of care tests AND evaluation). For full details of search terms please see S1 Table. We report our review following the Preferred Reporting Items for Systematic Reviews and Meta-Analyses (PRISMA) guidelines [37] (S2 Table).

Results were extracted into EndNote (Clarivate Analytics, Philadelphia, USA), and duplicates removed. Titles and abstracts were screened by EC according to the inclusion and exclusion criteria (Table 1). Articles that met the criteria, or any that were unclear, were included for full text review by all authors independently, with any discrepancies discussed as a group to reach consensus for final inclusion. SSF and EMHE independently extracted data from eligible articles into custom-made Excel (v2019, Microsoft) tables. References of included papers were also hand-searched by EC and new potentially eligible articles full-text screened by all authors before confirming inclusion, with data independently extracted by SSF and EMHE.

Study quality was assessed by SSF and EMHE, independently, using the Critical Appraisal tools for use in JBI Systematic Reviews where possible ([38] as recommended by [39]; S3–S7 Tables). For studies where the CT/NG POCT was implemented as routine (e.g. service evaluations), we modified the JBI checklist for analytical cross-sectional studies by removing the three questions relating to exposure and confounding, as no JBI checklists were appropriate. For before-after studies, the NHLBI quality assessment tool was used [40]. For questionnaire-based studies, the Center for Evidence-Based Management "Critical appraisal of a survey" checklist was used ([41] as recommended by [42]). Any differences between the two reviewers were reconciled through discussion to provide an overall study quality score calculated as number of questions with a "yes" response divided by the total number of questions. Any questions that were non-applicable were removed from the denominator.

We did not produce a protocol or register this study.

## Results

The systematic review search returned 440 articles, of which 26 were included for review. After the references of the 26 included articles were checked to confirm completeness, two further articles were eligible, which led to a final inclusion of 28 articles (Fig 1). Study quality assessment indicated that 5 studies were of low quality (≤50% criteria met), 4 studies were of medium quality (between 50 and 75% of criteria met) and the remaining 19 articles were of high quality (≥75% criteria met) (S3–S9 Tables).

The binx health io CT/NG has been implemented in a small number of clinical settings, however, available reports show the test being implemented in research-use only scenarios in the USA [43, 44] and publications (to-date of this review) reporting implementation in the UK

**Table 1. Inclusion and exclusion criteria.**

| | Inclusion | Exclusion |
|---|---|---|
| **Population** | • Humans | • Non-humans |
| **Intervention** | • Point of care or rapid tests Tests for combined genital chlamydia and gonorrhoea detection.<br>• Implemented as routine practice | • Tests that are not classed as point of care or rapid<br>• Tests that are not nucleic acid amplification tests<br>• Tests for infections other than genital chlamydia and gonorrhoea<br>• Tests that only detect chlamydia OR gonorrhoea<br>• Tests that are not *Conformité Européene* (CE)- or Food and Drug Association (FDA)- approved<br>• Tests not implemented as routine practice, e.g. implemented as a research-only tool |
| **Outcome** | • Evaluation of the implementation of the test as in routine practice (e.g. time to treatment) | • Research-only outcome (e.g. sensitivity and specificity; modelling of hypothetical scenarios) |
| **Type of study** | • Peer-reviewed primary research<br>• Conference/poster abstracts | • Grey literature<br>• Review articles<br>• Any other type of literature |
| **Date** | • Articles published up to 16/02/21 | • Articles published after 16/02/21 |
| **Language** | • Any | • Any |

do not report clinical and other health-related outcomes [45]. The Cepheid CT/NG GeneXpert has been implemented and evaluated for multiple outcomes measures in many settings around the world. As such, only the Cepheid GeneXpert platform, using the CT/NG dual diagnostic cartridge, was eligible for assessing compliance with international guidelines for CT/NG POCT development and evaluation, and evaluating the values placed on CT/NG POCTs implemented in routine practice in the published literature.

TPP and REASSURED criteria provide checklists to guide the development and evaluation of STI POCTs. A summary of both these frameworks for CT/NG POCTs is presented below

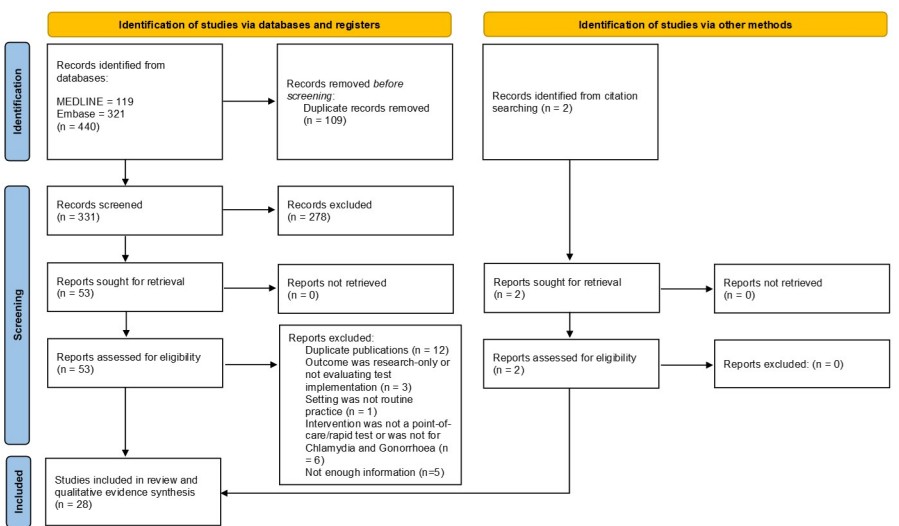

**Fig 1. PRISMA flowchart** [37]. *From*: Page MJ, McKenzie JE, Bossuyt PM, Boutron I, Hoffmann TC, Mulrow CD, et al. The PRISMA 2020 statement: an updated guideline for reporting systematic reviews. BMJ 2021;372:n71. doi: 10.1136/bmj.n71. For more information, visit: http://www.prisma-statement.org/.

**Table 2. The Cepheid CT/NG GeneXpert fulfilment of TPPs and REASSURED criteria.**

| Characteristic* | | | |
|---|---|---|---|
| Sensitivity/specificity genital samples (a,b) | Sample type | CT % sensitivity / specificity | NG % sensitivity / specificity |
| | Male Urine | 97.5 / 99.9 | 98.0 / 99.9 |
| | Female endocervical swab | 97.4 / 99.6 | 100.0 / 100 |
| | Female vaginal swab | 98.7 / 99.4 | 100.0 / 99.9 |
| | Female Urine | 97.6 / 99.8 | 95.6 / 99.9 |
| Pooled percent agreement extra-genital samples (a,b) | Sample type | CT % agreement positive / negative | NG % agreement positive / negative |
| | Rectal | 89.72% / 99.23% | 92.75% / 99.75% |
| | Pharyngeal | 89.96% / 99.62% | 92.51% / 98.56% |
| Use setting (a,b) | Table-top, not portable | | |
| | Level 2 service (district hospital) | | |
| Specimen (a,b) | Female and male urine, endocervical swab, vaginal swab, rectal swab and pharyngeal swab from asymptomatic and symptomatic patients | | |
| Steps; user-friendly (a,b) | ~4; sample preparation automated. Three-step process from sample provision to processing; sealed cartridge system; <1 minute hands-on time | | |
| Time to result (run-time) (a,b) | ~90 mins | | |
| Cold chain; reagent stability (a,b) | No; to be determined | | |
| Power (a,b) | Mains power or solar power | | |
| Training; user-friendly (a,b) | Less than half a day | | |
| Connectivity for monitoring, surveillance & data export (a,b) | Yes; computer/internet required; remote calibration; C360 platform system provides systems and epidemiology monitoring. Connectivity between GeneXpert and electronic patient records used to deliver results in published service evaluation [47]. | | |
| Equipment price (USD); per test price (a,b) | ~17,000 USD (with 4 modules), but could be higher; 16.20 USD (CT/NG) | | |
| Environmentally friendly (a,b) | No: single use cartridge; disposal of used materials via local medical waste regulations | | |
| Environmental tolerance of packaged test kit and operating conditions (robust—tolerance to difficult environmental conditions) (a,b) | Stable temperature and power required but has been used successfully in remote healthcare settings (see Table 3) | | |
| Internal quality control (a) | Yes. Sample Adequacy Control on each cartridge for increased results integrity | | |
| Sample capacity/through-put (a) | Various capacity readers available (single to 80 cartridge units); readers stackable for scale-up | | |

*a = TPP; b = REASSURED [8, 11, 12, 46–51].

(Table 2); these reflect a summary of published evaluations of the Cepheid CT/NG GeneXpert diagnostic performance.

## TPP recommendations for CT/NG POCTs

WHO TPP recommendations for CT/NG POCTs include: **sensitivity** (NG: 90% minimal, 98% optimal; CT: 90% minimal, 100% optimal) and **specificity** (NG and CT: 98% minimal, 100% optimal); **training requirements** (<90 minutes minimal, <30 minutes optimal); **time-to-results** (<60 minutes minimal, <30 minutes optimal) and **price per test** (<5 USD minimal, <1 USD optimal) [12]. Other considerations include the inclusion of a Global Positioning System (GPS) on the platform reader, and sample capacity/through-put [12]. Operational use prioritisation is suggested to be in the following order: ease of use, training, high tolerance to

difficult environmental conditions and long shelf life, self-contained quality control, data capture/connectivity/data export, biosafety and waste disposal [12].

## REASSURED criteria

The REASSURED criteria are: **Real-time connectivity** (feedback for patient treatment and connection to surveillance systems); **Ease of specimen collection** and **Environmentally-friendly** (non-invasive specimen collection; use of recyclable materials and reduction of hazardous waste); **Affordable** (<10.00 USD for a molecular assay); **Sensitive** (minimising false negatives) and **Specific** (minimising false positives); **User-friendly** (2–3 steps and minimal training required); **Rapid and robust** (15–60 minutes from sample-in to answer-out; withstands various weather and environmental conditions without refrigeration); **Equipment-free** (or utilises batteries or solar power) and **Deliverable to end-users** (ensures it reaches LMIC users) [11].

Published diagnostic evaluations show that the Cepheid CT/NG GeneXpert fulfils some, but not all, TPP and REASSURED criteria (Table 2). Minimal sensitivity and specificity requirements are met for genital samples, but not for extra-genital samples for CT. Training may be considered too long at "less than half a day", compared with the minimal TPP requirement for <90 minutes. The 90-minute time-to-results is longer than the 60-minute minimum of both criteria frameworks, and the cost is higher than the 10 USD recommendation for the REASSURED criteria. In addition, the test is not environmentally friendly, as it uses a single-use cartridge to be disposed of via local medical waste regulations. However, the test can be used with non-invasive specimen types, does feature connectivity for monitoring, surveillance and data export, and is user-friendly with automated sample preparation and a three-step process from sample provision to processing. It also features a sample adequacy control for internal quality control purposes. The REASSURED criteria "deliverable to end-users" can be considered contextually, such as the tests' compatibility with diagnostics currently in use (e.g. current use of the GeneXpert platform for tuberculosis or TV testing [46]), which relate to potentials to use pre-existing procurement and test supply chains in those settings.

Data from eligible articles were extracted to show the value of implementing the Cepheid CT/NG GeneXpert in three different healthcare service settings (specialist sexual health services, General Practice [GP] and other non-sexual health specialist services, and outreach services), spanning different income settings (Table 3).

Implementation of the Cepheid CT/NG GeneXpert demonstrated that faster (and appropriate) treatment was achieved in all settings. This was facilitated by reduced time to notification of results, which was a specific outcome for some studies [47, 58, 60, 63, 69] but same-day treatment was hindered by patients not waiting for test results at the point of care [61, 65], and one study specifically reported increased patient waiting time in-clinic [57]. However, implementation of the test was broadly acceptable in all settings reporting this as an outcome [55, 61, 66–68, 72]. In non-sexual health services, introduction of the Cepheid CT/NG GeneXpert enabled detection of STIs, a service which previously had not been available [57, 64, 66, 67, 69]. Additional benefits beyond immediate patient management were also recorded, including improving partner treatment, reducing transmission, and cost-savings [47, 55, 65, 69].

In Table 4, in addition to the Cepheid CT/NG GeneXpert studies included in Table 3, we report on a wider literature of stakeholder values for implementation of POCTs, including reports of the binx health io CT/NG. Studies reporting implementation of the binx CT/NG in a sexual health specialist service and a University student health clinic (outreach service) in the USA were restricted to implementation without results delivery; at the time of the studies the test was not yet FDA approved [43, 44, 73]. Publication of a UK-based project tracing

**Table 3. Routine implementation of the Cepheid CT/NG GeneXpert in different healthcare service settings.**

| Care setting | Study design and location | Target population | Test Implementation | Impacts assessed | Results |
|---|---|---|---|---|---|
| *Sexual health services* | GeneXpert test implementation and time to treatment comparison with same target population. San Francisco City STI Clinic, USA. May-Dec 2018. Cohen et al. 2019. | Asymptomatic MSM and transwomen attending follow-up care for HIV PrEP; those who were sexual contacts of someone with CT/NG were excluded | GeneXpert implementation as standard of care among MSM and transwomen | 1. Mean and median time to treatment | 1. 90 patients were NG/CT positive. After introduction of POCT, mean and median time to NG/CT treatment decreased from 6 and 4 days to 1.7 and 0 days (p<0.001) |
| | Comparison of standard care with "sample-first" (prior to consultation) pathway and use of in-house GeneXpert testing on patient management. Courtyard Clinic, St George's University Hospitals NHS Foundation Trust, London UK. Harding-Esch et al. 2017. | Males and females symptomatic for CT/NG infection; sexual contacts of CT/NG positive patients | Standard triage procedure followed by self-collected sample provided by patients prior to clinical consultation. GeneXpert testing, routine culture and microscopy, and non-NAAT POCTs for TV and BV. Results provided to patients in clinical consultation | 1. Proportion of patients who responded in favour of the 'sample first' approach<br>2. Proportion of patients who received test results in the same clinical visit<br>3. Individual durations of patient pathway component parts (triage, recruitment, sample collection, clinician consultation, discharge from clinic) | 1. 95.8% (23/24) of patients found self-sample provision prior to clinical consultation acceptable; no patients refused<br>2. 78.6% (55/70) of patients did not wait for POCT results before leaving clinic<br>3. GeneXpert results (CT/NG negative) led to: two females avoiding presumptive treatment; one male receiving treatment for possible *Mycoplasma genitalium* infection |
| | Implementation of GeneXpert in specialist sexual health clinic symptomatic service in London, UK. No dates given. Mandlik et al. 2017. | Subset of 100 symptomatic patients diagnosed with CT/NG | GeneXpert implementation as standard of care | 1. Time from attendance to treatment<br>2. Modelled number of partners not exposed due to earlier CT/NG treatment of index patient | 1. Time to treatment reduced by 66% (6.2 days) from 9.5 days pre-implementation to 3.3 days post-implementation.<br>2. 54% fewer partners were exposed to CT/NG (19.9 pre-implementation and 9.12 post-implementation) |
| | Retrospective review of patients' notes in sexual health clinic after GeneXpert introduction, London, UK. No dates given. Whitlock et al. 2015. | Patients diagnosed with CT/NG | Service redesign involving express screening service, including sexual history on touchscreen computers, self-collected samples, POC testing and automated results management | 1. Time to treatment | 1. Of 431 CT and/or NG diagnoses, time to treatment reduced by 190 hours |
| | Comparison of data between Dean Street Express (DSE; a walk-in, rapid STI screening service for asymptomatic individuals) and 56 Dean Street (56DS; standard off-site laboratory-based NAAT testing), London, UK, in one-year period from 1 June 2014 to 31 May 2015. Whitlock et al. 2018. | Patients attending DSE and 56DS. Data extracted from patient notes of first 12 patients (MSW, MSM and women) | GeneXpert implementation as standard of care at DSE. Sexual history is provided by patients on a touchscreen computer, which orders the relevant swabs based on self-reported sexual history. Patients self-collect swabs/samples, which are delivered to and processed on-site GeneXpert. Health adviser reviews sexual history, collects blood for off-site syphilis, HIV and/or hepatitis B/C testing (results within 4 hours). Treatment for test-positive patients is provided at 56DS | 1. Time from sample collection to notification of GeneXpert test results<br>2. Modelled reduction in partner notification and treatment (for partners exposed to CT/NG)<br>3. Modelled cost to clinics of fewer attendances for screening and treatment of partners<br>4. Modelled potential public health impact due to transmissions averted | 1. Time from sample collection to notification of GeneXpert test results for 138,261 test notifications reduced by 8.68 days between 56DS [8.95 days (95% CI 8.91–8.99 days)] and DSE [0.27 days (95% CI 0.26–0.28 days)]<br>2. Estimated 854 partner attendances averted<br>3. Estimated annual savings of £124,283 (IQR £4260–590,331) due to reduced partner attendances<br>4. Estimated 196 CT and/or NG transmissions averted |
| | Comparison of standard care and use of in-house GeneXpert testing and results notification pathway. Dean Street Express clinic, Chelsea and Westminster Hospital NHS Foundation Trust, London UK. 19 April 2013–7 January 2014. Wingrove et al. 2014. | Males and females asymptomatic for CT/NG infection | GeneXpert introduced into clinic for on-site testing | 1. Median time from testing to treatment for positive patients<br>2. Median time to informing of results for negative and positive patients<br>3. Median time from informing positive patients of results to treatment | 1. Median time from testing to treatment for patients with positive results (n = 28) was 2 days (IQR 1–6 days) with GeneXpert and 10 days (IQR 7–11 days) for standard care<br>2. Median time to informing of negative results (n = 50) was 1 day (IQR 1–2 days) for GeneXpert and 12 days (IQR 8–14 days) for standard care. Median time from testing to patient result delivery for positive patients was 1 day (IQR 1–3 days) with GeneXpert and 8 days (IQR 7–9 days) for standard care<br>3. Median time from informing patients to treatment was 1 day (IQR 0–2 days) with GeneXpert and 1 day (IQR 1–4 days) for standard care |

*(Continued)*

**Table 3.** (Continued)

| Care setting | Study design and location | Target population | Test Implementation | Impacts assessed | Results |
|---|---|---|---|---|---|
| *General Practice and other non-sexual health specialist services* | Assessment of introducing newly available STI POCTs and treatment. Alotau, Milne Bay Province, Papua New Guinea. August—December 2014. Badman et al. 2016 | Females ≥18 years attending their first clinic antenatal visit | Face to face interview with nurse: demographic and sexual behaviour data collection. Routine antenatal and provider-initiated HIV (Alere Determine HIV1/2) and syphilis (SD Bioline anti TP 3.0) screening via rapid test; Syphilis rapid test followed by confirmatory laboratory test. Self-collected vaginal swabs with on-site testing for CT/NG and TV (Cepheid GeneXpert) and BV (BV Blue). Positive patients (as needed): same-day antibiotic treatment; risk-reduction counselling; contact tracing | 1. Impact of introducing newly available STI POCTs on patient waiting times<br>2. Proportion of patients receiving same-day results<br>3. Proportion of participants positive for infection<br>4. Time to treatment; Proportion of positive patients receiving same-day treatment<br>5. Reasons for leaving prior to treatment | 1. Integration of study procedures into routine clinic activities resulted in an average of two hours' additional waiting time per patient<br>2. All participants (n = 125) received same-day results<br>3. 53.6% (67/125) of patients had CT, NG, TV or BV; of these 71.6% (48/67) were asymptomatic<br>4. Of those with an infection, 83.6% (56/67) received same-day treatment. All received treatment within one week.<br>5. Reasons for leaving prior to treatment included family commitments, and the need to travel significant distances back to their homes by foot or by bus |
| | Assessment of introducing GeneXpert into two university hospital family planning clinics: Antoine Béclère Hospital (Clamart, France) and Avicenne Hospital (Bobigny, France), July 2012—Jan 2013. Bourgeois-Nicolaos et al. 2015. | Women presenting to the clinics for induced abortion, intrauterine device insertion as emergency contraception, or signs of STI, were consecutively recruited | Patient samples sent for GeneXpert testing in hospital's laboratory. Test results reported to clinic by phone and/or fax. Patients with positive results were immediately telephoned and prescription faxed to their closest pharmacy. Prescriptions for partners, or letter to partner's physician, provided | 1. Test success rate<br>2. Proportion of patients receiving CT/NG result before termination procedure | 1. The rate of GeneXpert assay success was 98.3% (581/591) test success (not intermediate/invalid result) on first attempt<br>2. 100% of patients received appropriate treatment pre-termination procedure, compared with 40% with standard NAAT |
| | Assessment of GeneXpert implementation in Haitian Study Group for Kaposi's sarcoma and Opportunistic Infections (GHESKIO) clinics. GHESKIO provides "integrated primary care services, including HIV counselling, AIDS care, antenatal care, and management of tuberculosis and STIs." Port-au-Prince, Haiti, 26 Oct 2015–14 Jan 2016. Bristow et al. 2017. | Pregnant women ≥18 years attending GHESKIO clinics | Participants self-collected samples, which were tested by GeneXpert as standard of care. Women returned to GHESKIO within 7 days to receive test results and treatment if test-positive | 1. Proportion of patients consenting to participate (acceptability)<br>2. Proportion of infections treated (feasibility) | 1. 300/322 (93.2%) women consented to testing<br>2. 122/133 (91.7%) infections were treated |
| | Assessment of GeneXpert test implementation in Prince Cyril Zulu Communicable Disease Centre (PCZCDC), a large public healthcare clinic that provides "general primary health care services for adults free of charge" in Durban city centre, KwaZulu-Natal, South Africa, May 2016—Jan 2017. Garrett et al. 2018. | HIV-negative women, at high HIV risk, aged 18–40 years, attending PCZCDC for STI care | Implementation of GeneXpert, *Trichomonas vaginalis* (TV) (OSOM® Rapid Trichomonas Test), and bacterial vaginosis (BV) (Gram stain microscopy) to evaluate how expedited partner therapy (EPT) introduction could be accelerated through use of POCTs. Results available within 2 hours. Test-positive women were immediately treated and offered EPT packs. STI-positive women invited to participate in focus group discussions on POC testing and EPT. An EPT questionnaire was administered by telephone at one-week follow-up. Women were retested for STIs in the clinic after 6 and 12 weeks. | 1. Proportion of STI-positive women accepting EPT<br>2. Proportion of STI-positive women successfully delivering EPT at one-week FU<br>3. Acceptability and feasibility | 1. 62/63 (98.4%) women with an STI were offered EPT, and 54/62 87.1% accepted<br>2. At telephonic follow-up one week later, 48/54 (88.9%) reported successfully delivering EPT to partner at one-week follow-up (77.8% [42/54] observed); (11.1% [6/54] unobserved)<br>3. In focus group discussions, women (n = 29) reported being in favour of the new care model acceptable and supported the care model because "they received a rapid, specific diagnosis, and could facilitate their partners' treatment" |
| | Randomised controlled trial in an urban academic emergency department (ED), USA. April 2015—May 2016. Gaydos et al. 2019. | Women undergoing pelvic examinations and CT/NG testing as part of their ED standard of care | Control: standard-of-care CT/NG NAAT, with 2- to 3-day turnaround time. Intervention: rapid GeneXpert test, in addition to the standard-of-care NAAT. Rapid results immediately provided, and treatment provided to all patients according to providers' clinical judgment | 1. Proportion of patients under-treated<br>2. Proportion of patients over-treated<br>3. Length of stay | 1. Undertreatment for CT/NG was 0% for the intervention group (0/10 & 10/5) and 43.8% (6/13 and 4/7) for the control group<br>2. Clinicians unnecessarily provided treatment for CT in 46.5% (53/114) of uninfected control group participants compared with 23.1% (27/117) of intervention group participants. Clinicians unnecessarily provided treatment for NG in 46.7% (56/120) of control group participants compared with 25.4% (31/122) of intervention group participants<br>3. The length of stay did not differ significantly between groups |

(*Continued*)

**Table 3.** (Continued)

| Care setting | Study design and location | Target population | Test Implementation | Impacts assessed | Results |
|---|---|---|---|---|---|
| | Cross-over cluster randomised controlled trial of routine GeneXpert implementation to improve infection management (intervention; n = 6 health services) compared to standard care (control; n = 6 health services). Primary health services that provide care to Indigenous people in regional or remote locations in Western Australia, Far North Queensland, and South Australia. June 1, 2013—Feb 29, 2016. Guy et al. 2018. | Patients aged 16–29 years attending participating health services in a 12-month period | Health services were provided training for use of the GeneXpert and equipment, supplies for ≤150 GeneXpert tests; participating services were reimbursed for retesting (Further details provided [52]) | 1. Primary outcome: proportion of patients found to have CT or NG who had a positive result at retesting 3 weeks to 3 months after treatment<br>2. Secondary outcomes:<br>3. Proportion of infections treated within 7 days of sample collection date;<br>4. Proportion of patients who were given treatment on the same day as testing<br>5. Proportion of patients who were given treatment within 2 days of testing<br>6. Proportion of patients who were given treatment within 7 days of testing<br>7. Proportion of patients who were given any treatment within 4 months of a positive CT/NG test result<br>8. Staff acceptability<br>9. Patient acceptability | 1. Proportion of positive-test individuals retested between 3 weeks and 3 months after treatment was: 14% (63/455) in intervention group (19%; 12/63 had positive retest result) versus 17% (67/405) in control group (13%; 9/67 had a positive retest result)<br>2. Of all individuals with a positive test in the intervention group, 76% (347/455) were treated within 7 days compared with 47% (191/405) in the control group (absolute across-cluster difference of 29%)<br>3. In the intervention group, 49% (221/455) were given treatment on the same day as testing compared with 27% (111/405) in the control group<br>4. In the intervention group 60% (274/455) of patients were treated within 2 days, compared with 30% (122/405) in the control group<br>5. In the intervention group 76% (347/455) of patients were treated within 7 days, compared with 47% (191/405) in the control group<br>6. In the intervention group 94% (427/455) of patients were treated within 4 months, compared with 86% (347/405) in the control group<br>7. Clinical staff (N = 35) found GeneXpert testing highly acceptable<br>8. Patient acceptability surveys (N = 80) indicated a high degree of satisfaction with GeneXpert testing |
| | Randomised controlled trial in an urban ED, Washington DC, USA, Oct 2013—Oct 2014. May et al. 2016. | Symptomatic patients presenting to an urban ED, and where treating provider was ordering diagnostic CT/NG test | Control: standard-of-care CT/NG NAAT, with results available within 1–4 days Intervention: rapid GeneXpert test, with results provided during ED visit. Treatment was provided at ED provider's discretion. After patient discharge, treating physician filled out a clinician survey | 1. Antibiotic overtreatment rates<br>2. Treatment adherence<br>3. Symptom resolution 7 to 10 days post-discharge<br>4. Results notification<br>5. Healthcare utilisation and charges, and total ED charges | 1. Clinicians unnecessarily provided treatment for CT and/or NG to 11/20 (55.0%) control group participants, compared with 8/37 (21.6%) intervention group participants (P = 0.01)<br>2. intervention group participants were less likely to report missed antibiotic doses (Risk Difference [RD], −51.3%; 95% CI, −84.4% to −18.2%; Risk Ratio [RR], 0.23; 95% CI, 0.06–0.88)<br>3. No differences were found in symptom resolution 7 to 10 days post-discharge between intervention and control group participants<br>4. Intervention group participants were more likely to be notified of their results (RD, 50.6%; 95% CI, 22.7%–78.5%; RR, 2.72; 95% CI, 1.26–5.86)<br>5. There were no significant differences in healthcare charges or utilisation, or total ED charges |
| | Assessment of GeneXpert introduction in antenatal clinic (ANC), Kinshasa, Kisantu health zone, Democratic Republic of Congo. No dates given. Mvumbi et al. 2017. | Pregnant women attending ANC | Trained clinic staff collected observed if women presented with STI symptoms, and collected vaginal swabs. Samples were tested using GeneXpert CT/NG and TV tests | 1. Proportion symptomatic patients STI-positive<br>2. Proportion asymptomatic patients STI-positive | 1. 10/352 (2.8%) women were symptomatic; 5/10 (50%) were CT/NG/TV positive<br>2. 50/342 (14.6%) asymptomatic patients were CT/NG/TV positive |
| | Comparison of patients tested with GeneXpert C to a historical control group tested using a traditional NAAT in an urban community teaching hospital ED, Dec 2014–Jan 2015. Rivard et al. 2017. | Patients ≥15 years of age who were tested for NG/CT | GeneXpert implementation as standard of care. Test-positive patients who received results prior to ED discharge were provided with notification, counselling, and treatment on-site. For patients whose results were not available pre-discharge, providers could offer empiric treatment and then follow-up with results post- discharge | 1. Percentage of patients who received appropriate initial treatment during their index ED visit (test-positive patients receiving antimicrobial therapy in concordance with the CDC guidelines and test-negative patients not receiving antimicrobial treatment)<br>2. Factors independently associated with appropriate treatment<br>3. Time to test results<br>4. Time to patient notification of positive test results<br>5. Time to appropriate treatment<br>6. Cost of appropriate and inappropriate treatment | 200 consecutive patients tested by GeneXpert compared with 200 historical patients tested with traditional NAAT.<br>1. 60% of patients received appropriate initial treatment in the historical group, compared with 72.5% in the GeneXpert group (P = 0.008). This was predominantly due to avoiding unnecessary treatment test-negative patients<br>2. CT/NG test availability prior to discharge was the only factor associated with appropriate treatment (odds ratio [OR], 22.65 [95%CI, 2.86–179.68, P = 0.003])<br>3. Median time to test results was 2.4 hours (1.4–12.0) in the GeneXpert group compared with 31.7 hours (9.7–105.9) in the historical group (P<0.001)<br>4. Median time to patient notification of positive test results was 17.4 hours (0.0–93.0) in the GeneXpert group compared with 53.7 hours (26.9–79.9) in the historical group (P = 0.010)<br>5. Mean time to appropriate treatment for test-positive patients was 4.9 ± 21.3 hours in the GeneXpert group compared with 23.0 ± 56.3 hours in the historical group<br>6. GeneXpert testing cost $343,566 over the study duration compared with $348,457 in the historical group, saving $4891 ($24.46 per patient) |

(*Continued*)

**Table 3.** (Continued)

| Care setting | Study design and location | Target population | Test Implementation | Impacts assessed | Results |
|---|---|---|---|---|---|
| | Assessment of GeneXpert implementation in Princess Marina Hospital ANC (the main government referral hospital for southern Botswana), Gaborone, Botswana, July—October 2015. Wynn et al. 2016. | Women receiving antenatal care at the clinic, who were aged ≥18 years, gestational age <35 weeks, mentally competent and willing to return to clinic for follow-up care | Women self-collected vaginal swabs, which were tested on-site in the ANC vitals room by GeneXpert for CT, NG, and TV. Women received same-day test results notification, in-person or by telephone. Test-positive women received same-day treatment prior was to leaving clinic | 1. Acceptability of intervention<br>2. Feasibility of intervention<br>3. Treatment uptake | 1. 200/225 (89%) eligible women accepted to participate.<br>2. 100% of eligible women were successfully tested for CT, NG and TV, and received same-day results. 143 (72%) women received results in-person prior to leaving clinic, and 57 (29%) were contacted by telephone after leaving the clinic (6 [10.5%] of these were test-positive and returned to clinic for treatment)<br>3. 100% of test-positive women were successfully treated, 80% immediately |
| | Assessment of GeneXpert introduction in one main clinic and three sex-on-premises venues (SOPV) where regular outreach HIV/syphilis POC testing had been taking place, within an urban community context, Brisbane, Australia, 3 March 2017–14 June 2018. Bell et al. 2020. | Prospective consecutive sampling of asymptomatic patients (predominantly MSM), ≥16 years, presenting at any of the four included locations. Patients reporting potential HIV exposure within the past 72 hours of attendance were excluded | Pilot of peer-delivered, community-led service providing POC CT/NG testing. GeneXpert implementation as standard of care in included settings. Participants self-collected samples, which were tested by GeneXpert at main clinic. Participants received their CT/NG results by telephone or SMS within 24 h. Test-positive participants referred for treatment, either in-clinic or elsewhere (community-based services, sexual health services, regular GP and non-regular GP). Peer test facilitators conducted follow-up telephone interviews with test-positive participants 2 weeks post-referral for retesting and treatment. Additional online 'Post-Referral Survey' for test-positive participants at 2-week post-testing follow-up interview phone call. | 1. Acceptability and feasibility<br>2. Time to results notification<br>3. Proportion of treated patients<br>4. Proportion of contacts<br>5. Estimated additional number of CT/NG infections detected | 1. CT/NG POCT accepted on 93.4% (4523/4843) occasions; 99.3% of patients accepted on their first visit. Uptake varied by setting: 93.8% (4051/4318) at clinic vs. 89.9% (472/525) from the three SOPVs combined (P<0.001). Post-Referral Survey and Evaluation Survey results indicated patients found the service acceptable, accessible, and would recommend the service.<br>2. 604/614 (98.4%) test-positive participants received their result and were referred for treatment within 24 h of testing. Ten (1.6%) were 'lost to follow up'<br>3. 89.7% (70/78) of participants reported receiving treatment<br>4. Post-referral, 64.1% (50/78) of participants reported informing all their contacts<br>5. Estimated 117 CT and 66 NG infections would not have been identified if the service was not offering CT/NG testing |
| *Outreach services* | Assessment of GeneXpert implementation in a mobile healthcare van at an annual community event in a metropolitan area with high STI prevalence. 2012 and 2013, no specific location given. Hesse et al. 2015. | Males and females ≤14 years | All specimens were self-collected in the van. Participants with positive results were notified and prescribed treatment. Questionnaire to assess acceptability of test turnaround times and self-sample collection | 1. Treatment delivery rates<br>2. Patient acceptability of testing | 1. 2/12 (16.6%) females and 0/10 (0%) males were CT positive and none were NG positive using GeneXpert testing. 1/2 (50%) positive patients was notified of her results and received same-day treatment<br>2. 30 participants (20 females; 10 males) completed the questionnaire. Sample collection was as acceptable in a van as in the doctors' office; faster turn-around-times for STI testing results were considered the most acceptable |
| | Assessment of GeneXpert introduction and same-day CT/NG treatment. May 2017 to June 2019, Los Angeles California and New Orleans Louisiana, USA. Keizur et al. 2020. | Young people ages 12–24 years with high sexual risk behaviours, recruited online and self-advertisements in homeless shelters, lesbian, gay, bisexual, and transgender organizations and community health centres in Los Angeles, California, and New Orleans, Louisiana USA | Every 4 months, within a 24-month enrolment period, participants attended clinic and self-collected pharyngeal, rectal, and urine or vaginal samples for CT/NG testing using GeneXpert. Positive patient management: Before March 2018 in Los Angeles and November 2018 in New Orleans: participants were referred to a local clinic or their primary care doctor for treatment. After March 2018 in Los Angeles and between 12 November 2018 and 28 February 2019 in New Orleans: participants were offered same-day treatment and expedited partner therapy packs by study staff | 1. Proportion of participants who received same-day treatment<br>2. Participants' median time to treatment<br>3. Number of partner treatment packs taken by participants<br>4. Any reported adverse treatment-related events | 1. The proportion of participants receiving same-day CT and NG treatment increased from 3.6% (5/140) pre-intervention to 21.1% (20/95) post-intervention<br>2. Median time to treatment decreased from 18.5 days pre-intervention to 3 days post-intervention<br>3. 37.9% (n = 36) participants took a median of 1 partner treatment pack each (range 1–3; 48 total)<br>4. No reported treatment-related adverse events |
| | Assessment of GeneXpert implementation in four community-based settings in Harare, Zimbabwe, participating in CHIEDZA trial (Community based interventions to improve HIV outcomes in youth), June 2019—Jan 2020. Martin et al. 2021. | All youth, aged 16–24 years, accessing CHIEDZA services. | GeneXpert testing within 48 hours of first-catch urine sample provision. Participants able to collect test result the following week, with positive-test participants actively followed-up. | 1. STI testing uptake<br>2. Proportion of test-positive participants treated<br>3. Proportion of test-positive participants symptomatic.<br>4. Contacts traced and treated<br>5. Factors associated with testing uptake | 1. Uptake was 33·3% (1478/4440; 95% CI 31·9–34·7); 30·4% (294/967) in men and 34·1% (1184/3473) in women<br>2. 67% (165/248) test-positive participants treated<br>3. 3% (7/248) test-positive participants symptomatic and received syndromic management<br>4. 87/248 (35.1%) partners attended for treatment<br>5. Current STI symptoms were independently associated with testing uptake. Uptake also motivated by potential to be treated if positive, and perceived risk based on their own or partner's sexual behaviour. Stigma and lack of confidentiality were barriers to testing. |

(*Continued*)

**Table 3.** (Continued)

| Care setting | Study design and location | Target population | Test Implementation | Impacts assessed | Results |
|---|---|---|---|---|---|
| | Assessment of GeneXpert implementation in urban Walk In Ruhr (WIR) inter-institutional care centre, Germany, Dec 2016 – July 2018. Skaletz-Rorowski et al. 2020. | Asymptomatic youth (14–30 years) approached in schools, universities and youth centres attending sexual health education lectures; sample collection took place at WIR inter-institutional care centre | GeneXpert platform implemented within WIR centre. Samples tested by nurses or doctors immediately after collection | Turn around time (TAT) was defined as the interval between when the swabs were provided to the patient to the time communication of the result to the patient. 1. Median turnaround time (TAT) (time between swab provision and patients receiving results) 2. Time between test and starting treatment he interval between initiation of test to initiation of therapy was additionally documented. | 272 participants (133 males, 133 females). 1. Median TAT was 3:09 hours; 91.8% received their positive test result within 24 hours, and 95.7% within 48 hours. This compares with standard TAT of 72 hours 2. Median time between test and starting treatment was 6:50 hours; 73.3% received initial treatment within 24 hours, and 86.7% within 48 hours. This compares with standard time to treatment of approximately 120 hours 73.3% with a positive result received initial treatment within 24 h and 86.7% within 48 h |

[47, 52–72].

implementation into routine care did not include assessment of clinical outcomes [73]. Nevertheless, the available studies (albeit limited to the USA and UK) have shown implementation processes of this CT/NG POCT to be highly acceptable to patients [43, 44] and healthcare workers [44, 45, 73].

Some of the values for molecular CT/NG POCTs cross-cut all settings: unmet need, care pathway context, and accountability. However, the majority of values are setting specific. Sexual health services and outreach services have the least overlap in values, whereas GP and other non-sexual health specialist services "bridge" between them. GP and other non-sexual health specialist services and outreach services share the value that the test is most likely to be used as a screening tool to increase testing, rather than the multiple purposes of screening, diagnosis, and guiding use of treatment, as is necessary in sexual health services. Non-specialist settings also have similarities for evidence of cost-effectiveness and translation challenges as this often requires new staff and training [52, 74, 75]. Sexual health services and GP and other non-specialist services overlap most for change in practice and change in resource requirement, and implementation metrics.

## Discussion

The Cepheid CT/NG GeneXpert test was the only molecular CT/NG POCT to have been implemented and evaluated as part of routine practice in the published literature. Although it did not meet all TPP or REASSURED criteria, review of its implementation and reported benefits demonstrated this did not preclude it from bringing value to a service or its stakeholders. Of note, although the cost-per-test of the Cepheid CT/NG GeneXpert exceeds the minimum TPP and REASSURED recommendations [49], cost-effectiveness models show relative value-for-money of POCTs when considering onward transmission and progression of disease [50, 75–78]. Thus, it is only when these tools and frameworks are examined within the delivery context that sense-making happens around the adoption and implementation decision-making processes [26, 79]. This was further emphasised when extracting these findings for development of the value proposition, where we found that values differed both within and between healthcare settings.

To our knowledge, this is the first report that systematically reviews the literature on molecular CT/NG POCTs' implementation in routine practice, to assess the value different stakeholders in different settings place on them. Furthermore, we have synthesised this evidence to

Table 4. Value proposition for molecular CT/NG POCTs by setting type, based on the value proposition for laboratory medicine [35].

| Value proposition | Sexual health services | General Practice and other non-sexual health specialist services | Outreach services |
|---|---|---|---|
| **Healthcare context** (Is it a service redesign issue? Is it a quality improvement issue? Are there any potential conflicts of interest between stakeholders, e.g. disinvestment in other stakeholders' resources, e.g. alternative diagnostic technology?) | Specialist health service (level 3) Impact on existing resources and contracts must be considered. | Primary health service (level 1). If added to pre-existing services, impact on existing resources and contracts must be considered. | Community health service (level 0). If added to pre-existing services, impact on existing resources and contracts must be considered. |
| **Unmet need** (Is it a clinical, process, and/or economic problem?) | Unmet need is likely to be context and stakeholder dependent. Faster results delivery, reduced time to treatment and appropriate patient management (clinical and process). Potential for increased timely access to sexual healthcare. Patient acquisition of results within one visit has potential to improve CT/NG diagnosis process for patients and healthcare professionals (i.e. via reduction of patient recall). | | |
| **Care pathway context** (Is it a screening, diagnosis, or monitoring issue?) | Screening and diagnosis: in general CT/NG tests are used either for diagnostic purposes in individual clinical cases, or for national screening programmes. As these are not chronic conditions repeat performance of tests to monitor CT and NG over time is not needed. However, test of cure is recommended for all patients diagnosed with NG, and for some patients with CT. | | |
| **Test and its utility(ies)** (Is it for screening, diagnosis, candidacy for treatment, guiding use of treatment, monitoring efficacy of, and compliance with, treatment?) | Screening, diagnosis, guiding use of treatment. | Can be used as a **screening** tool to increase testing opportunistically among asymptomatic patients and enables symptomatic patients access to rapid **diagnosis** and **guides treatment** when needed. | Can be used as a **screening** tool to increase testing among asymptomatic populations with a high prevalence of infection and enables access to rapid **diagnosis** and **guides treatment** when needed. |
| **Resource requirement** (What will be the cost of the test? Will there be additional resource requirement, or redundancy, in other parts of the organization?) | Costs of tests is likely to be higher than laboratory tests, though cost savings may be made in reduction of staff costs for patient recall. Reduction of time for healthcare professionals to conduct patient recall. May necessitate changes to clinical pathways / duration of patient visit to accommodate test time to results to enable same-day results delivery and treatment if needed. | Costs of tests is likely to be higher than laboratory tests, though costs savings may be found in reducing inappropriate antibiotic treatment. May necessitate changes to clinical pathways / duration of patient visit to accommodate test time to results to enable same-day results delivery and treatment if needed. | Costs of tests is likely to be higher than laboratory tests. Redeployment of healthcare professionals may need to be employed to enable outreach service. Mobile testing van, new or existing community space will be needed to provide testing and treatment. |
| **Benefits of using test** (Will it improve diagnosis and treatment, process of care, and/or patient experience? Will it reduce cost of care?) | Faster time to results improves faster time to treatment where needed which may result in: reduction in inappropriate treatment (reduced syndromic treatment); expedited partner therapy; reduction in onward progression of disease (sequelae). | Faster time to results improves faster time to treatment where needed, which may result in: avoiding unnecessary treatment; reduction in loss to follow-up and recall efforts; expedited contact tracing; onward progression of disease (sequelae); potential for widening testing and screening coverage. | Faster time to results improves faster time to treatment where needed, which may also result in: reduction in inappropriate treatment (reduced syndromic treatment); reduction of onward progression of disease (sequelae); expedited access to treatment for contacts; potential for widening testing and screening coverage. |
| **Impact on outcomes** (Will it improve patient morbidity and mortality, access to care, and/or efficiency of care? Will it reduce the complications of care?) | Potential to increase appropriate antibiotic treatment for infections. Potential for reduced time to results and treatment to reduce unnecessary antibiotic use. Reduction in inappropriate treatment (reduced syndromic treatment) may reduce antibiotic resistance. | Potential to raise awareness among healthcare professionals and thus increase their offer of STI testing to patients. Potential to increase appropriate antibiotic treatment for infections. Potential for reduced time to results and treatment to reduce unnecessary antibiotic use. Improvement in patient access to STI testing enables earlier treatment of previously undiagnosed infections. | Improvement in patient access to STI testing enables earlier treatment of previously undiagnosed infections. Potential to increase appropriate antibiotic treatment for infections. |

(*Continued*)

**Table 4.** (Continued)

| Value proposition | Sexual health services | General Practice and other non-sexual health specialist services | Outreach services |
|---|---|---|---|
| **Evidence of clinical effectiveness** (Is there evidence of improved diagnostic accuracy? Is there is evidence of improved clinical outcome?) | Evidence of similar accuracy to laboratory-based tests must be established. Improved clinical outcomes including reduction in inappropriate treatment (reduced syndromic treatment); onward progression of disease (sequelae). | | |
| **Evidence of cost effectiveness** Is there evidence of cost effectiveness when using the test?) | May reduce costs to the clinic and reduce healthcare professional time; cost-effectiveness models in higher-income countries show value-for-money when considering transmission and progression of disease. | Cost effectiveness is likely to depend on impact of diagnoses on larger public health outcomes (as per specialist service modelling). | Cost effectiveness is likely to depend on impact of diagnoses on larger public health outcomes (as per specialist service modelling). |
| **Translation challenges** (What is the plan for translating the evidence of effectiveness into routine practice?) | The instrument should be easy to use and allow connectivity to existing clinical recording systems to provide rapid access to results. Guidance for implementation of new tests in services is often lacking; clinical leads are responsible for overseeing clinical pathway changes so implementation is likely to be service-driven and thus inconsistently delivered. | The instrument should be easy to use and allow connectivity to existing clinical recording systems to provide rapid access to results. Training in equipment may be needed prior to implementation. Training in sexual healthcare provision may be needed for healthcare professionals. | The instrument should be easy to use and allow connectivity to existing clinical recording systems to provide rapid access to results. Training in equipment may be needed prior to implementation. Training in sexual healthcare provision may be needed for healthcare professionals. |
| **Change in practice** (Will there be a revised care guideline, e.g. revised diagnostic pathway) | Stakeholder engagement is necessary to enable implementation. May necessitate changes to clinical pathways / duration of patient visit to accommodate test time to results to enable same-day results delivery and treatment if needed. | Stakeholder engagement is necessary to enable implementation. May necessitate changes to clinical pathways / duration of patient visit to accommodate test time to results to enable same-day results delivery and treatment if needed. | Stakeholder engagement is necessary to enable implementation. Provision of STI / CT/NG screening where previously none present. |
| **Change in process** (Will there be rapid access to results, reduction in clinic visit requirement, care provided in different setting?) | Reduction in time to result. Rapid access to infection-specific treatment (for CT and/or NG positive patients). | Reduction in time to result. Reduction in follow-up visits for those found positive for infection. | Provision of STI / CT/NG screening where previously none present. |
| **Change in resource requirement** (Will there be reduced use of alternative diagnostic tools, reduced length of stay, reduced need for hospitalization?) | If time to results cannot be achieved within the standard clinical visit time, patients will have an increased length of stay. The number of patients managed with POCTs may result in the reduction of laboratory-based CT/NG tests conducted. | If time to results cannot be achieved within the standard clinical visit time, patients will have an increased length of stay. The number of patients managed with POCTs may result in the reduction of laboratory-based CT/NG tests conducted. | Additional resources may be needed to provide this as a new service. |

(*Continued*)

**Table 4.** (Continued)

| Value proposition | Sexual health services | General Practice and other non-sexual health specialist services | Outreach services |
|---|---|---|---|
| **Implementation metrics** (What are the intermediate outcome measures (clinical, process and economic), e.g., HbA1c, new test usage, previous test usage, time to treatment, clinic visits, length of stay, to be employed in performance management of implementation) | **Clinical outcome measures**: number of patients appropriately treated; number of partner notifications averted; number of patient follow-up visits averted; number of patients receiving same-day result; numbers of partners appropriately treated. **Process outcome measures**: Feasibility and acceptability among healthcare professionals and patients; time from sample taking to result to patient; impact on patient waiting times (as compared to standard care). **Economic outcome measures**: initial costs and ongoing cost of POCT contract, as compared with standard care (laboratory-based testing); clinical pathway change cost comparison, i.e., reduction of treatment, follow up and contact tracing costs, any change to staff time for testing and results delivery (as per specialist services). | **Clinical outcome measures**: number of patients appropriately treated; number of patients receiving same-day result; number of partners appropriately treated. **Process outcome measures**: Feasibility and acceptability among healthcare professionals and patients; time from sample taking to result to patient; impact on patient waiting times (as compared to standard care). **Economic outcome measures**: initial costs and ongoing cost of POCT contract, as compared with standard care (laboratory-based testing); clinical pathway change cost comparison, i.e., reduction of follow up and contact tracing costs, any change to staff time for testing and results delivery. | **Clinical outcome measures**: number of patients appropriately treated; number of patients receiving same-day result; number of partners treated. **Process outcome measures**: Feasibility and acceptability among healthcare professionals and patients; time from sample taking to result to patient. **Economic outcome measures**: cost per screening; cost per infection detected; total cost of service. |
| **Accountability** (Who will benefit from use of test? Who may experience dis-benefit? Who will manage the implementation?) | There is potential for benefit to the health system, health care professionals, patients and the population. | | |
| | Healthcare professionals will be responsible for performing the tests and managing new patient care pathways. | | |
| | Potential disbenefit to population infection surveillance systems. | | |
| | The number of patients managed with POCTs may result in the reduction of laboratory-based CT/NG tests conducted. | | |

develop a value proposition to facilitate decision-making around their integration into sexual healthcare. By reviewing the Cepheid CT/NG GeneXpert's implementation, we were able to demonstrate the diversity of use in various healthcare settings (specialist, non-specialist, and outreach), and in different areas of the world, allowing a more robust review of the test's value from multiple stakeholder perspectives.

Of particular interest is the finding that sexual health specialist and outreach services had the least overlap in values. This underlines the need for specific measures of value to be identified by service type: if a service does not already exist (as in outreach), mapping outcome measures such as costs of changes to existing clinical pathways and associated costs of task redeployment are redundant, whereas these are clearly important measures if redesigning an existing sexual health clinic service. Similarly, consideration of existing services provided within specific settings matter: measuring the impact of replacing traditional laboratory CT/NG NAATs with POCTs (including impact on current laboratory contracts) requires different evaluation indicators than does the replacement of syndromic management of possible infections with POCTs. The examples presented here highlight our key finding: the value of novel diagnostic test adoption and implementation is perceived differently depending on your setting and stakeholder role. It is therefore critical for the values for the specific service to be identified before a test and its mode of implementation are chosen.

We did not consider studies that focused solely on test performance, or reports on test implementation in research-use-only environments, as we considered these outside the remit of this report. By limiting ourselves to implementation studies, we may have missed identifying additional values of the test, although we tried to address this by including research study outcomes in the final value proposition (Table 4). We only searched two databases (Medline and

Embase), but given the subject area, the inclusion of conference abstracts, and the fact we searched references of included papers, we think all relevant publications have been identified. It is likely, however, that there are other cases of implementation that have not been reported in the literature; publications available will most likely reflect the values of the study authors, and not those of all stakeholders involved. Furthermore, multiple value propositions for POCTs exist [34, 35]; we chose one, based on its relevance to laboratory medicine and inclusion of diversity of stakeholder values.

We found variability between reports of implementation studies and their outcomes, as well as in their study quality. Not all of the studies included reported on clinical pathways (i.e. procedures), which would have outlined how the test was used. As a result, we cannot directly compare the results of each setting, which precluded metanalysis and limits our ability to understand the value of the test to each stakeholder in each of the different contexts. We encourage authors to have clear objectives, to report on outcomes matching these objectives, and to follow the appropriate international standards of reporting for their chosen study design. However, although uniformity would enable better evaluation across different settings, it would unlikely reflect the diversity of outcomes that need to be measured in those different settings; the heterogeneity in study design, test implementation and impacts assessed in the literature in itself demonstrates the variability of values placed on molecular CT/NG POCTs by different stakeholders and in different settings. For example, the inclusion of qualitative studies in the value proposition we propose enabled us to broaden our understanding of the contextual values of these POCTs. We suggest more work be done to understand the values of a wider variety of stakeholders in order to encourage them to be actively involved in study design and implementation, which would lead to reporting of more relevant outcomes of interest. We also encourage reflexive reporting on lessons learned, particularly with regards to study design and outcomes measures; if any data were found to be important when assessing the POCTs for adoption but were not thought to be important when the evaluation was designed, this would be useful to consider in future studies and their design.

Despite the diversity of Cepheid CT/NG GeneXpert implementation mechanisms, there were commonalities among study outcomes to explore. Patient benefit was measured in each, although the indicators that were measured varied, including numbers of patients who received a same-day result, time between clinic visit and/or sample taking and result provision, and patient acceptability. Among CT/NG positive patients, time to treatment, and partner notification/treatment measures were also commonly reported.

Healthcare professionals are a particularly important stakeholder group for implementation and previous research to identify an ideal test has focused on them [80]; clinicians are often responsible for new pathway construction [52, 81], and research has shown that nurses' inclusion in quality improvement projects may improve job satisfaction and reduce workforce instability [82]. Healthcare providers across the included studies placed value in patient benefits, specifically the reduced time to result notification, and for those patients testing positive, reduced time to treatment. Qualitative studies, in particular those among healthcare professionals participating in the TTANGO studies in Australia, reported high levels of satisfaction with the Cepheid CT/NG GeneXpert, and related this to their belief in the test's ability to improve patient and public health outcomes, as well as the device's ease of use [83, 84]. However, in some studies [61, 65], the faster time to results delivery was negatively impacted by patients being unwilling or unable to wait for their results at the point of care; qualitative studies providing insight into the appropriate implementation of CT/NG POCTs into routine healthcare practice may help to mitigate this issue [85].

No test fulfils all the WHO TPP or (RE)ASSURED criteria [9, 11, 12]. However, even less-than-perfect technologies have the potential to improve patient outcomes [86, 87]; waiting for

the ideal molecular POCT before implementing tests that are currently available has implications both for individual patients and public health [88, 89]. This research synthesis shows the potential for less-than-perfect CT/NG POCTs to hold value for multiple stakeholders in different healthcare settings. Therefore, we recommend that stakeholders in sexual healthcare explore the potential for existing POCTs to provide value to their services. As more molecular CT/NG POCTs are developed and approved by regulatory bodies, the specific characteristics of each may be more or less suited to particular settings, and the value proposition developed could help decision-makers determine the most important values for them and their stakeholders to guide test choice.

## Conclusions

Criteria have been set for the development of ideal CT/NG POCTs. Similarly, guidance has been developed for the adoption of novel diagnostics into health systems. This guidance is necessary to protect patients and direct health systems towards efficient use of resources to meet public health goals, and attempts to cater to a diverse range of stakeholder needs and expectations. The plurality of these needs means that a single test is unlikely to be viewed as a panacea or "magic bullet" for solving the clinical, social and structural issues around provision of CT/NG diagnosis across all settings. Stakeholders wishing to improve their service through the implementation of CT/NG POCTs should be supported to identify the values most relevant to their settings and context rather than waiting for the ideal test to be produced: there is no magic bullet.

## Supporting information

**S1 Table. Full list of search terms.**
(DOCX)

**S2 Table. PRISMA flow diagram.**
(DOCX)

**S3 Table. Quality assessment for all included articles, by article, article type, score and percentage and location in manuscript tables.**
(DOCX)

**S4 Table. Service evaluation article assessment.**
(DOCX)

**S5 Table. Before and after study article assessment.**
(DOCX)

**S6 Table. Randomised controlled trial study article assessment.**
(DOCX)

**S7 Table. Qualitative study article assessment.**
(DOCX)

**S8 Table. Questionnaire study article assessment.**
(DOCX)

**S9 Table. Economic study article assessment.**
(DOCX)

## Acknowledgments

Special thanks to Professors Christopher Price and Trisha Greenhalgh for organising the *Maximising Value from New Diagnostic Tests* workshop in 2019, which inspired this publication.

## Author Contributions

**Conceptualization:** Sebastian S. Fuller.

**Data curation:** Sebastian S. Fuller, Emma M. Harding-Esch.

**Formal analysis:** Sebastian S. Fuller, Emma M. Harding-Esch.

**Investigation:** Sebastian S. Fuller, Eleanor Clarke, Emma M. Harding-Esch.

**Methodology:** Sebastian S. Fuller, Emma M. Harding-Esch.

**Validation:** Sebastian S. Fuller, Emma M. Harding-Esch.

**Writing – original draft:** Sebastian S. Fuller, Emma M. Harding-Esch.

**Writing – review & editing:** Sebastian S. Fuller, Eleanor Clarke, Emma M. Harding-Esch.

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
