## [Decision Letter · Decision Letter 0]

9 Sep 2021

PONE-D-21-17643Molecular chlamydia and gonorrhoea point of care tests implemented into routine practice: systematic review and value proposition developmentPLOS ONE

Dear Dr. Fuller,

Thank you for submitting your manuscript to PLOS ONE. After careful consideration, we feel that it has merit but does not fully meet PLOS ONE’s publication criteria as it currently stands. Therefore, we invite you to submit a revised version of the manuscript that addresses the points raised during the review process.

We look forward to receiving your revised manuscript.

Kind regards,

Catherine E Oldenburg

Academic Editor

PLOS ONE

Journal Requirements:

2. Thank you for stating the following in the Competing Interests/Financial Disclosure* (delete as necessary) section:

“We have read the journal's policy and the authors of this manuscript have the following competing interests: ADREU has received funding from Abbott, binx health, Cepheid, SpeedDx, Mologic, Revolugen and Sekisui, for the research and evaluation of their diagnostics. SSF was a co-investigator on Innovate UK grant to binx health, "A stratified medicine diagnostic test for STI patients at the point-of-care" (ref: 971543) and is a consultant for the WHO on point-of-care tests for sexually transmitted infections. EHE and EC report that no competing interests exist.”

We note that one or more of the authors are employed by a commercial company: binx health, WHO

3. Please include your tables as part of your main manuscript and remove the individual files. Please note that supplementary tables (should remain/ be uploaded) as separate "supporting information" files

Reviewers' comments:

Reviewer's Responses to Questions

**Comments to the Author**

1. Is the manuscript technically sound, and do the data support the conclusions?

Reviewer #1: Yes

Reviewer #2: Partly

2. Has the statistical analysis been performed appropriately and rigorously? 

Reviewer #1: N/A

Reviewer #2: N/A

3. Have the authors made all data underlying the findings in their manuscript fully available?

Reviewer #1: Yes

Reviewer #2: Yes

4. Is the manuscript presented in an intelligible fashion and written in standard English?

Reviewer #1: Yes

Reviewer #2: No

5. Review Comments to the Author

Reviewer #1: he article is a systematic review of published literature on the implementation of point-of-care tests (POCTs) for Chlamydia trachomatis (CT) and Neisseria gonorrhoeae (NG) infections into clinical settings. The paper is well-written and helps to consolidate research findings across many different clinical settings (STI clinics, general practice, and community-based outreach) and in a variety of global health contexts (high-income and low- and middle-income settings). The systematic review identified 28 relevant articles on implementation of CT/NG POCTs, all of which were using the Gene Xpert platform. The device did not meet all criteria set forth by target product profiles or REASSURED, but it met many on those two lists. The findings show how the implementation of CT/NG POCTs depend on the local context and the different stakeholders in various settings. Reporting on implementation outcomes varies across studies, which makes it difficult to compare, but the authors provide detailed data on the various outcomes measured and how they might impact different settings. The discussion is appropriate to the study and data presented. Overall, a well-done article on a very interesting topic.

There are a few suggestions that would improve the manuscript.

1. The authors do a good job of describing their methods for screening manuscripts and data extraction. However, it is not clear how they conducted their search. They could provide their search terms and methodology, which would make it more clear to the readers.

2. There were 278 articles identified and after screening, 252 were excluded. The authors provide the inclusion and exclusion criteria in Table 1, but it would also be helpful to understand how many articles were excluded for those reasons. Perhaps they could provide percentage for each of the categories listed in Table 1. The authors could provide this information in the Fig 1 flow diagram exclusion box. This would help the reader understand the exclusion process.

3. While the discussion is appropriate, it would be strengthened by providing some additional context to readers about how the results and the synthesis of evidence that the authors completed in Table 4 could be used by different decision makers. For example, in paragraph 5 of the discussion (lines 325-328), the authors discuss their key finding - how the adoption and implementation of tests depend on the setting and stakeholder roles, which are critical to identify before implementation begins. Perhaps the authors could give some examples or scenarios to illustrate this point and to make the use a bit more clear. Similarly, in paragraph 2 of the discussion (lines 296-302), the authors also make an effort to highlight their development of value propositions to help decision-making; this section might also benefit from being more explicit in how they envision decision makers using their value propositions. Perhaps these two paragraphs could be combined to give additional context and to make these important findings more clear.

4. In lines 360-362, the authors encourage researchers implementing POCTs for CT/NG to report their results. Do they authors have any specific guidance or recommendations on conducting those future research studies (eg - what to measure, how to measure, outcomes to report, etc). That information would be very helpful considering one limitation the authors noted was the heterogeneity of measured outcomes between studies, which made it difficult to compare.

5. Recommend avoiding the use of RHS as an acronym, as it is not a common acronym and is only used three times.

6. In Table 3: Could consider labeling Column “Outcomes assessed” instead of “Impacts Assessed”

7. Table 4: In Impact on Outcomes row, it seems that “Potential to increase appropriate antibiotic treatment for infections” and “Potential for reduced time to results and treatment to reduce unnecessary antibiotic” would also be relevant for sexual health services and outreach services columns. But they are not listed in those columns.

8. Small grammatical issues:

- Line 284 typo “NT” instead of “NG”

- In abstract, would say “One author” instead of EC, as it was not immediately clear by reading the abstract that this was one of the authors.

- Line 70 – recommend not starting sentence with a digit

Reviewer #2: 1. This manuscript looks more like a narrative review to describe some interesting points from the literatures on POCTs for CT and NG.

2. The paper is based on published studies and conference abstracts, but the strategy used to search the literatures are not quite clear for me although it is known two databases were used.

3. The authors mentioned that search strategy was formed to answer several proposed research questions. What strategy was used and was different question based on different search strategy?

4. It is not quite clear if these included studies are those studies on POCT evaluation or those on POCT implementation or both? The purposes of these two types of studies are different. For example, the results shown in Table 2 (The Cepheid CT/NG GeneXpert fulfils some, but not all, TPP and REASSURED criteria) seem to address the findings from evaluation studies while Table 3 is on implementation. There have been a few published systematic reviews and meta-analyses on evaluation findings of molecular POCTs.

5. As the authors mentioned, the (RE)ASSURED and TPP criteria are proposed to guide the R&D of POCT-based diagnostics. The question of “Do molecular CT/NG POCTs implemented in routine practice fulfil the (RE)ASSURED and TPP criteria?” seems more relevant to POCT characteristic or performance issue rather than implementation one. For example, the element E (equipment-free) in (RE)ASSURED, it is not possible that POCT does require an equipment and in routine practice the equipment is actually not required.

6. For example, “Implementation of the Cepheid CT/NG GeneXpert demonstrated that faster (and appropriate) treatment was achieved in all settings” – this is actually relevant to one characteristic (getting results in 90 mins) of Cepheid CT/NG GeneXpert. “Faster treatment was achieved” does only mean the local adherence to the POCT process to initiate the treatment in 90 mins.

6. PLOS authors have the option to publish the peer review history of their article (what does this mean?). If published, this will include your full peer review and any attached files.

Reviewer #1: No

Reviewer #2: No

---

## [Author Response · Author response to Decision Letter 0]

20 Oct 2021

“Molecular chlamydia and gonorrhoea point of care tests implemented into routine practice: systematic review and value proposition development”

Response to reviewers

We thank the reviewers for their time and helpful comments, which have helped improve the clarity of the manuscript and allowed us the opportunity to expand some thoughts further. Line numbers correspond to those in the “clean” version.

Reviewer 1: 

The article is a systematic review of published literature on the implementation of point-of-care tests (POCTs) for Chlamydia trachomatis (CT) and Neisseria gonorrhoeae (NG) infections into clinical settings. The paper is well-written and helps to consolidate research findings across many different clinical settings (STI clinics, general practice, and community-based outreach) and in a variety of global health contexts (high-income and low- and middle-income settings). The systematic review identified 28 relevant articles on implementation of CT/NG POCTs, all of which were using the Gene Xpert platform. The device did not meet all criteria set forth by target product profiles or REASSURED, but it met many on those two lists. The findings show how the implementation of CT/NG POCTs depend on the local context and the different stakeholders in various settings. Reporting on implementation outcomes varies across studies, which makes it difficult to compare, but the authors provide detailed data on the various outcomes measured and how they might impact different settings. The discussion is appropriate to the study and data presented. Overall, a well-done article on a very interesting topic.

Many thanks for this summary of the article and your positive comments.

1. The authors do a good job of describing their methods for screening manuscripts and data extraction. However, it is not clear how they conducted their search. They could provide their search terms and methodology, which would make it more clear to the readers.

We have clarified our methods section by adding further details of our search strategy in the text, and clearly sign-posting where all details on our search strategy can be found in supplemental materials (lines 142-170 methods).

2. There were 278 articles identified and after screening, 252 were excluded. The authors provide the inclusion and exclusion criteria in Table 1, but it would also be helpful to understand how many articles were excluded for those reasons. Perhaps they could provide percentage for each of the categories listed in Table 1. The authors could provide this information in the Fig 1 flow diagram exclusion box. This would help the reader understand the exclusion process.

We are unsure where these figures are arising from. As indicated in the PRISMA flowchart (Figure 1), initially 440 articles were returned, of which, 109 were duplicates and so removed, leaving a total of 331 articles initially identified. Of these, 278 were excluded in the first round (title and abstract screening) as they did not meet eligibility requirements. Of the 53 articles that were screened for full text, 26 were found eligible; a further 2 articles were added following our reference search. We report the specific reasons for exclusion of the 25 articles that were screened for full text, in figure 1, where our findings are inputted into the PRISMA template for systematic review reporting. 

3. While the discussion is appropriate, it would be strengthened by providing some additional context to readers about how the results and the synthesis of evidence that the authors completed in Table 4 could be used by different decision makers. For example, in paragraph 5 of the discussion (lines 325-328), the authors discuss their key finding - how the adoption and implementation of tests depend on the setting and stakeholder roles, which are critical to identify before implementation begins. Perhaps the authors could give some examples or scenarios to illustrate this point and to make the use a bit more clear. Similarly, in paragraph 2 of the discussion (lines 296-302), the authors also make an effort to highlight their development of value propositions to help decision-making; this section might also benefit from being more explicit in how they envision decision makers using their value propositions. Perhaps these two paragraphs could be combined to give additional context and to make these important findings more clear.

Thank you for this suggestion to strengthen our discussion. We have added to and rearranged our discussion, including combining the two paragraphs as suggested by the reviewer, to highlight how different settings and contexts considered for CT/NG POCT use have highlighted differing values of the test in lines 310-321.

4. In lines 360-362, the authors encourage researchers implementing POCTs for CT/NG to report their results. Do they authors have any specific guidance or recommendations on conducting those future research studies (eg - what to measure, how to measure, outcomes to report, etc). That information would be very helpful considering one limitation the authors noted was the heterogeneity of measured outcomes between studies, which made it difficult to compare.

We found variability in reports of these implementation studies. Based on this, we encourage authors to have clear objects, and report on outcomes matching those objectives following the appropriate international standards of reporting for their chosen study type. However, although uniformity would enable better evaluation across different settings, it is unlikely to reflect the diversity of outcomes that need to be measured in those different settings. We suggest more work be done to understand the values of a wider variety of types of stakeholders in order to encourage them to be actively involved in study design and implementation, which would lead to reporting of more relevant outcomes of interest. We also encourage reporting on any lessons learned, particularly with regards to study design and outcomes measures; if any data were found to be important when assessing the POCTs for adoption but were not thought to be important when the evaluation was designed, this would be useful for future study design. We have now added these points to the discussion (lines 335 – 353).

5. Recommend avoiding the use of RHS as an acronym, as it is not a common acronym and is only used three times.

Thank you for calling this to our attention, we have removed this acronym.

6. In Table 3: Could consider labeling Column “Outcomes assessed” instead of “Impacts Assessed”

We agree and have changed this. 

7. Table 4: In Impact on Outcomes row, it seems that “Potential to increase appropriate antibiotic treatment for infections” and “Potential for reduced time to results and treatment to reduce unnecessary antibiotic” would also be relevant for sexual health services and outreach services columns. But they are not listed in those columns.

Thank you for pointing this out, we have reviewed the included manuscripts and data tables. We have corrected table 4and now included “Potential to increase appropriate antibiotic treatment for infections” and “Potential for reduced time to results and treatment to reduce unnecessary antibiotic” to the sexual health services column. We have also added “Potential to increase appropriate antibiotic treatment for infections” into the outreach services column. We did not find evidence of Potential for reduced time to results and treatment to reduce unnecessary antibiotic” among outreach services’ evaluations so have not included this. 

8. Small grammatical issues:

- Line 284 typo “NT” instead of “NG”

- In abstract, would say “One author” instead of EC, as it was not immediately clear by reading the abstract that this was one of the authors.

- Line 70 – recommend not starting sentence with a digit

Thank you for calling these to our attention, we have now made the recommended corrections. 

Reviewer 2:

1. This manuscript looks more like a narrative review to describe some interesting points from the literatures on POCTs for CT and NG.

This article is a systematic review and value proposition development, as is stated in the aims and described in the methods.

2. The paper is based on published studies and conference abstracts, but the strategy used to search the literatures are not quite clear for me although it is known two databases were used.

Please see our response here to a similar query to reviewer 1. 

3. The authors mentioned that search strategy was formed to answer several proposed research questions. What strategy was used and was different question based on different search strategy?

Please see our response here to a similar query to reviewer 1. 

4. It is not quite clear if these included studies are those studies on POCT evaluation or those on POCT implementation or both? The purposes of these two types of studies are different. For example, the results shown in Table 2 (The Cepheid CT/NG GeneXpert fulfils some, but not all, TPP and REASSURED criteria) seem to address the findings from evaluation studies while Table 3 is on implementation. There have been a few published systematic reviews and meta-analyses on evaluation findings of molecular POCTs.

Thank you for the opportunity to clarify the process.

As stated on lines 142-143 we conducted our systematic review on CT/NG POCT implementation.

As the reviewer has correctly seen, we have taken findings from the literature on POCT evaluation, in order to complete tables 1 & 2, which speak to the diagnostic characteristics of the Cepheid CT/NG GeneXpert and if these meet the current guidelines. We present these data to allow the reader to have access to all published information on the values of a CT/NG POCT in one document, (e.g. evaluation characteristics alongside our findings from the implementation literature). We have now clarified this in our article, lines 142-153 (methods) 197 – 204; 230 (results).

5. As the authors mentioned, the (RE)ASSURED and TPP criteria are proposed to guide the R&D of POCT-based diagnostics. The question of “Do molecular CT/NG POCTs implemented in routine practice fulfil the (RE)ASSURED and TPP criteria?” seems more relevant to POCT characteristic or performance issue rather than implementation one. For example, the element E (equipment-free) in (RE)ASSURED, it is not possible that POCT does require an equipment and in routine practice the equipment is actually not required.

We respectfully disagree with the reviewer on the usefulness of including CT/NG POCT performance characteristics in our article. The published literature on POCT acceptability to stakeholders shows that performance characteristics of the POCT are essential to the decision to implement (or not). As we have stated throughout the article, and is shown by previously published literature, multiple data points are used by stakeholders for the decision to implement. These data are not found anywhere in one place, and our article is intended to be a useful guide to potential implementors by presenting all relevant data that can be found in the literature in a synthesised format. 

6. For example, “Implementation of the Cepheid CT/NG GeneXpert demonstrated that faster (and appropriate) treatment was achieved in all settings” – this is actually relevant to one characteristic (getting results in 90 mins) of Cepheid CT/NG GeneXpert. “Faster treatment was achieved” does only mean the local adherence to the POCT process to initiate the treatment in 90 mins.

We believe this is a continuation of the previous point; please see our response above. Further, we disagree that ““Faster treatment was achieved” does only mean the local adherence to the POCT process to initiate the treatment in 90 mins.” The literature we have sourced shows that this is not always the case. 90 minutes is the test run-time, but time to treatment involves consideration of test implementation within the specific setting. Implementation of the POCT in several settings showed that initiation to treatment was longer than 90 minutes (and so not adhering to the “POCT process”), yet faster initiation of treatment was still achieved compared with the clinic’s previous diagnostic method. In contrast, in other settings, providing test results within 90-120 minutes did not always result in patients receiving results and treatment faster, if they were unwilling to wait even this amount of time. Thus, results are variable – which is a key point of this paper: the same test will be used differently and have different outcomes in different settings.

---

## [Editor Report · Decision Letter 1]

22 Oct 2021

Molecular chlamydia and gonorrhoea point of care tests implemented into routine practice: systematic review and value proposition development

PONE-D-21-17643R1

Dear Dr. Fuller,

We’re pleased to inform you that your manuscript has been judged scientifically suitable for publication and will be formally accepted for publication once it meets all outstanding technical requirements.

Kind regards,

Catherine E Oldenburg

Academic Editor

PLOS ONE
---

## [Editor Report · Acceptance letter]

29 Oct 2021

PONE-D-21-17643R1 

Molecular chlamydia and gonorrhoea point of care tests implemented into routine practice: systematic review and value proposition development 

Dear Dr. Fuller:

I'm pleased to inform you that your manuscript has been deemed suitable for publication in PLOS ONE. Congratulations! Your manuscript is now with our production department. 

Kind regards, 

on behalf of

Dr. Catherine E Oldenburg 

Academic Editor

PLOS ONE